# Fine-scale diversity of microbial communities due to satellite niches in boom and bust environments

**Yulia Fridman**[1☺], **Zihan Wang**[2,3☺], **Sergei Maslov**[2,3]*, **Akshit Goyal**[4]*

**1** National Research Center "Kurchatov Institute", Moscow, Russia, **2** Department of Physics, University of Illinois at Urbana-Champaign, Urbana, Illinois, United States of America, **3** Carl R. Woese Institute for Genomic Biology, University of Illinois at Urbana-Champaign, Urbana, Illinois, United States of America, **4** Physics of Living Systems, Department of Physics, Massachusetts Institute of Technology, Cambridge, Massachusetts, United States of America

☺ These authors contributed equally to this work.
* maslov@illinois.edu (SM); akshitg@mit.edu (AG)

**Data Availability Statement:** Code availability. All code comprised of custom Python scripts, using Python v3+, along with the Numpy v1.20.1 and Scipy v1.6.2 packages, and can be found at https://

## Abstract

Recent observations have revealed that closely related strains of the same microbial species can stably coexist in natural and laboratory settings subject to boom and bust dynamics and serial dilutions, respectively. However, the possible mechanisms enabling the coexistence of only a handful of strains, but not more, have thus far remained unknown. Here, using a consumer-resource model of microbial ecosystems, we propose that by differentiating along Monod parameters characterizing microbial growth rates in high and low nutrient conditions, strains can coexist in patterns similar to those observed. In our model, boom and bust environments create satellite niches due to resource concentrations varying in time. These satellite niches can be occupied by closely related strains, thereby enabling their coexistence. We demonstrate that this result is valid even in complex environments consisting of multiple resources and species. In these complex communities, each species partitions resources differently and creates separate sets of satellite niches for their own strains. While there is no theoretical limit to the number of coexisting strains, in our simulations, we always find between 1 and 3 strains coexisting, consistent with known experiments and observations.

## Author summary

Recent genomic data have revealed the remarkable spectrum of microbial diversity in natural communities surrounding us, which harbor not only hundreds of species, but also a handful of closely related strains within each of those species (termed "oligo-colonization"). While the mechanisms behind species coexistence are much better studied, the mechanisms behind the coexistence of closely related strains have remained understudied. Here, using a simple consumer-resource model, we show that if strains differ on their Monod growth parameters, they can coexist even on a single limiting resource, provided that the environments, specifically resource concentrations, vary with time in boom and

github.com/maslov-group/Coexistence_of_g_and_K. Data availability. All the numerical data from simulations can be found on the GitHub repository, at the following link: https://github.com/maslov-group/Coexistence_of_g_and_K. Source data are available at the GitHub repository, under the "data" directory. No new experimental data was generated during this study.

**Funding:** This research was supported in part by the National Science Foundation under Grant No. NSF PHY-1748958 and the Gordon and Betty Moore Foundation Grant No. 2919.02. A.G. is supported by the Gordon and Betty Moore Foundation as a Physics of Living Systems Fellow through Grant No. GBMF4513. The funders had no role in study design, data collection and analysis, decision to publish, or preparation of the manuscript.

**Competing interests:** The authors have declared that no competing interests exist.

bust cycles. The Monod growth parameters describe how a strain's growth rate changes with resource concentration, namely the half-maximal concentration and maximal growth rate. Simulations of our model show that both in simple and complex environments, even though an arbitrary number of strains can coexist, typically it is between 1 and 3 strains of a species that coexist over several randomly assembled communities, consistent with some experimental observations. This is because the allowed parameter space for coexistence shrinks significantly with the number of strains that coexist.

## Introduction

Microbial communities in almost all natural settings are characterized by an astonishing diversity, manifesting itself at multiple evolutionary scales [1, 2]. These scales range from separate domains (e.g., archaea and bacteria) all the way down to closely-related strains of the same species [3–7], other environments. This wide-ranging diversity can persist even in well-controlled laboratory settings, containing alternating cycles of exponential growth, followed by a transfer to fresh media after dilution by a large factor [8, 9]. The coexistence of distantly related community members, such as different species or kingdoms, can be readily explained via niche theory, which suggests that each species can occupy a different niche, e.g., by specializing on a different resource, allowing everyone to coexist [10–15]. Remarkably, it is the coexistence of fine-scale diversity—that is, closely related strains of the same species—that remains puzzling [7, 16, 17]. This is because presumably, such strains haven't diverged sufficiently to establish and occupy distinct resource niches. Thus, the observation of fine-scale diversity suggests that every resource might contain multiple niches: a primary niche and several "satellite" niches, ready to be occupied by closely related strains, enabling their coexistence. Here, the primary niche is occupied by a strain most competitive at high nutrient concentrations, and each "satellite" niche is occupied by a strain which is more competitive in a specific range of lower nutrient concentrations.

Satellite niches are not expected in stable environments, where resources are supplied continuously at a constant rate and reach a fixed concentration, such as in a chemostat. In these conditions, competitive exclusion guarantees that only the strain best adapted to the steady-state concentration will be able to survive [13]. In contrast, in fluctuating environments where nutrient concentrations change in time, the existence of satellite niches remains a possibility [18–20]. Indeed, in the extreme case of environmental fluctuations, i.e., in boom and bust cycles, where resource concentrations may vary over orders of magnitude, there is ample opportunity for several strains to coexist [21, 22]. Furthermore, strains of the same species may rapidly (with just a few mutations [23]) modify the range of nutrient concentrations optimal for their growth, allowing for rapid colonization of the available satellite niches by closely related strains, rather than by members from distant species. In conclusion, satellite niches may arise as a consequence of boom and bust cycles, and then be occupied by different strains of the same species, allowing their coexistence.

Here, we use a consumer-resource model of microbial communities to demonstrate how satellite niches appear and are colonized, over the process of community assembly. We start by considering the case of a single resource, and show that in this case, anywhere between 1 and 3 strains typically coexist. Indeed, the coexistence of even more strains is possible, but has a small likelihood. We then proceed to generalize our results to multiple resources (or niches), colonized by distantly-related species. In this case, each of the species comprises a small number of closely-related strains, which can coexist due to the presence of satellite niches. Our

results provide a possible mechanism for the widespread observation of fine-scale diversity in natural as well as laboratory microbial communities.

## Model and results

### Strains of the same species can coexist on a single limiting resource

To study whether multiple strains of a species—which differed in their maximal growth rate ($g$) and substrate affinity ($K$)—could coexist in boom and bust environments, we simulated a simple consumer-resource model inspired by serial dilution experiments. Briefly, both strains were initially inoculated in a medium consisting of one resource (e.g., carbon source) provided at a concentration $c_0$, allowed to grow for time period $T$, and then diluted by a factor $D$ and transferred to a fresh medium with resource at the same concentration $c_0$ (Fig 1B). We repeated these growth-dilution cycles until the community reached a steady state, that is, displaying reproducible dynamics in each cycle. Each strain with abundance $N_i$ grew exponentially, with its instantaneous growth rate determined by its Monod kinetics, its dynamics given as follows:

$$\frac{dN_i(t)}{dt} = g_i \frac{c(t)}{c(t) + K_i} N_i(t), \tag{1}$$

$$\frac{dc}{dt} = \sum_i -\frac{dN_i}{dt} = \sum_i -\frac{g_i\, c(t)}{c(t) + K_i} N_i(t) \tag{2}$$

where $c(t)$ represented the resource concentration at time $t$ after the start of each cycle ($c(t = 0)$ $= c_0$). Here, $g_i$ represented the maximal per capita growth rate of each strain, realized at high resource concentrations ($c(t) >> K_i$), and $K_i$ represented the resource concentration at which the growth rate of the strain dropped to half its maximum value. To simplify notation, we also assume here that the yields of all microbial species are equal to 1. We discuss the general case where the yields for different species is in the S1 Text. Note that the strain with a higher $g_i$ exhibits a fast initial growth, whereas a strain with a lower $K_i$ continues to grow at appreciable growth rates even at relatively low resource concentrations. Similar to previous results [22], we first considered a model with two strains of the same species, but to build on previous studies, we will later generalize this model to include multiple strains and species. In the two strain case we considered first, one of the strains (B in red) had a higher maximal growth rate $g_i$, analogous to an r-strategist in classical ecology [24, 25], whereas the other (A in blue) had a much lower value of $K_i$, analogous to a K-strategist in classical ecology (Fig 1B).

Simulations with two strains showed that both r and K strategists can coexist over a wide range of parameters $c_0$ and $D$ (Fig 1C), as observed in previous studies (e.g., Supplementary Fig 13A in ref. [26] and Fig 6 in ref. [22]). To quantify the range of environmental parameters in which both strains could coexist, one can measure the area of the space of parameters $c_0$ and $D$ where both strains survive (the region between the black lines in Fig 1C).

Alternatively, one can fix the environmental parameters $c_0$ and $D$ and quantify the range of growth parameters of strain B which would allow it to coexist with strain A with a given set of growth parameters, $g_A$ and $K_A$. Our analytical calculations show that the following inequalities need to be satisfied for the two strains A and B to coexist with each other (see S1 Text for

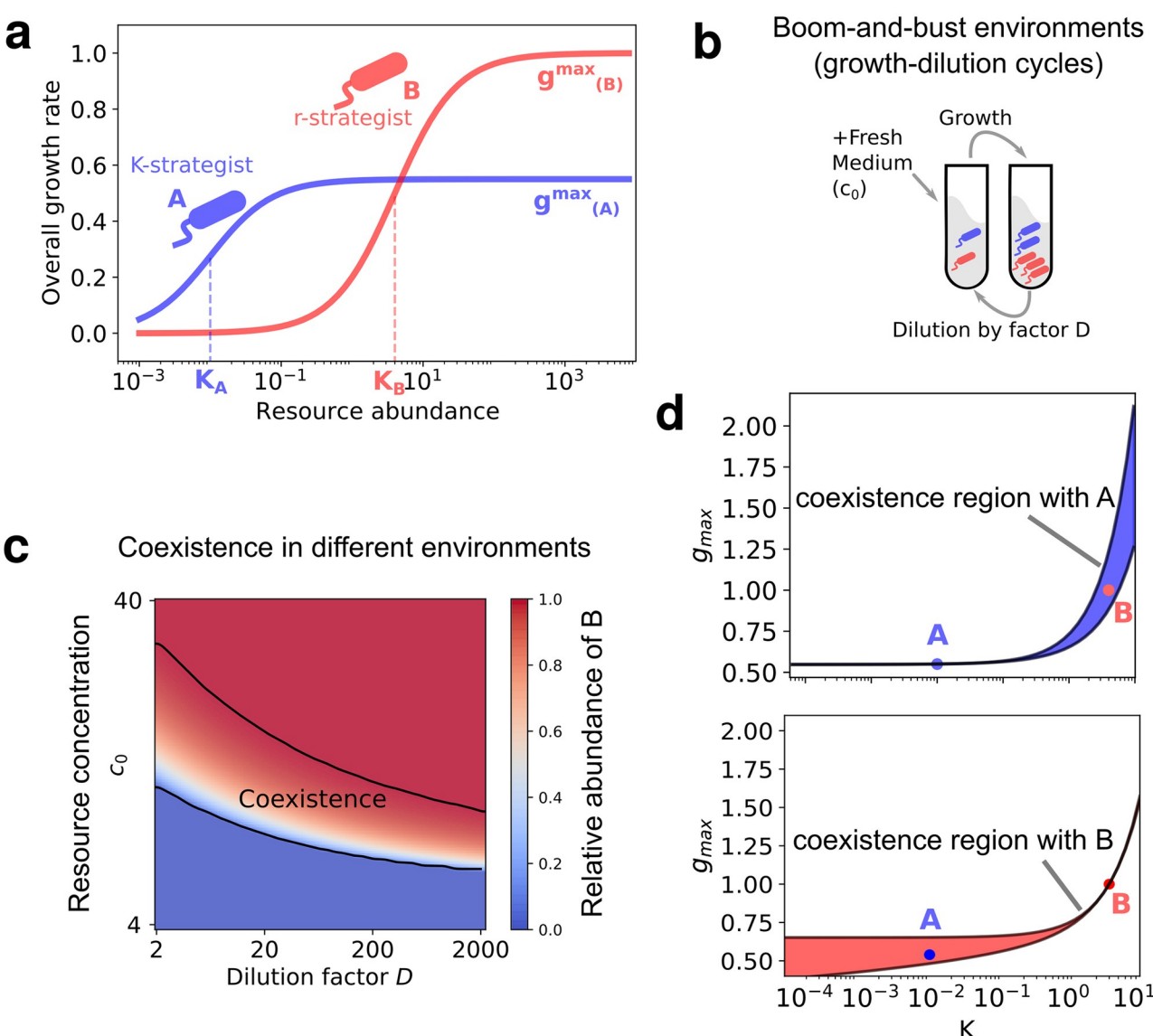

**Fig 1. Coexistence of two strains in a boom and bust environment supplied with a single limiting resource. (a)** Strains A and B follow Monod's law of growth. Strain A is a *K*-strategist with a lower maximum growth rate $g_{max}$ and substrate affinity($K$) compared with strain B, which is an *r*-strategist. **(b)** Schematic of the serial dilution experiment. At the end of each cycle, we dilute the population by a factor $D$, and supply fresh medium with resource concentration $c_0$ (see Methods). **(c)** Feasible environmental parameters for two strains with given growth rates. Each point on the heatmap defines an environment, characterized by $c_0$ and $D$, and the color of each cell represents the relative abundance of strain B at steady state. Black lines show the range of environments where both strains, A and B, coexist. **(d)** Coexistence region in terms of growth parameters of strain *B* for a fixed environment ($D = 10$, $c_0 = 10$) and fixed strain *A*. The blue region represents the range of growth parameters of B (red dot) where it can coexist with strain A (blue dot) in a fixed environment. Black solid lines are theoretical boundaries of the region.

complete derivation, which uses the idea of invasion fitness [27]):

$$\frac{(c_A^{(tot)} + K_B)\log D}{(c_A^{(tot)} + K_A)\log D + (K_A - K_B)\log \frac{c_0 + K_B}{c_A^{(f)} + K_B}} < \frac{g_B}{g_A} <$$

$$< \frac{(c_B^{(tot)} + K_B)\log D - (K_A - K_B)\log \frac{c_0 + K_A}{c_B^{(f)} + K_A}}{(c_B^{(tot)} + K_A)\log D}, \tag{3}$$

where $c_{A,B}^{(f)}$ is the concentration of nutrient left over at the end of each steady state growth cycle, and is defined by the following equation:

$$\frac{1}{g_{A,B}} \left( \log D + \frac{(D-1)\,K_{A,B}}{Dc_0 - c_{A,B}^{(f)}} \log \frac{Dc_0}{c_{A,B}^{(f)}} \right) = T, \tag{4}$$

and $c_{A,B}^{(tot)}$ is the total material concentration (cells and nutrients combined to have the same units) in each steady state growth cycle, and is defined by the following expression:

$$c_{A,B}^{(tot)} = \frac{Dc_0 - c_{A,B}^{(f)}}{D-1}. \tag{5}$$

In Fig 1D, we visualize (in blue) the range of coexistence given by Eq (3). It is extremely narrow when $K_B$ is below or slightly larger than $K_A$, but expands significantly as $K_B$ starts to increase compared with $K_A$. Our calculations show that the width of the region, $\Delta g_{\text{coexist}}$ bounded from above by $g_B^{(max)}$ and from below by $g_B^{(min)}$, for which coexistence is possible, can approximately be written as follows (see S1 Text):

$$\Delta g_{\text{coexist}} = g_B^{(max)} - g_B^{(min)} = g_A \cdot \frac{(K_B - K_A)^2}{K_A c_0} \cdot \frac{D-1}{D \log D}. \tag{6}$$

Here we assume that (i) $K_{A,B} \ll c_0$, (ii) $c_{A,B}^{(f)} \ll c_0$, achieved when growth cycles are sufficiently long so that nutrients are nearly fully depleted, i.e., $T \gg \log D/g_{A,B}$, and finally (iii) $(K_B - K_A) \ll K_{A,B}$. In a more general case where conditions (i) and (ii) are still valid, but (iii) is relaxed, one should replace $(K_B - K_A)^2/K_A$ with $(K_B - K_A) \log\left(\frac{K_B}{K_A}\right)$ (S1 Text). Note that the width of this region depends only mildly (roughly logarithmically for large $D$) on the dilution factor $D$ compared with its inverse linear dependence on the resource concentration $c_0$, suggesting that coexistence of strains should be possible for a broad range of environmental fluctuations, quantified by the dilution factor $D$.

In Fig 1e, we show the complementary range of coexistence for a strain $B$ (red in Fig 1e), which lay inside the coexistence range of strain $A$ (Fig 1D). Note that coexistence range of $B$ is broad for values of $g_A$ and $K_A$ below the growth parameters of $B$, and narrow on the opposite side ($g_A$ and $K_A$ above $g_B$ and $K_B$, respectively). This is in contrast with the coexistence range of $A$, which is instead broader on the right.

Thus, coexistence between two strains $A$ and $B$ is possible in this environment because $B$ can consume and grow on low concentrations of the available resource, on which strain $A$ grows relatively poorly. In such boom and bust environments, "satellite niches" appear, enabling coexistence between $A$ and other strains, which must lie in the region shown in Fig 1D. In the rest of this manuscript, we will refer to this coexistence region for a given strain as its "shadow".

## Coexistence of three or more strains

Now we discuss the circumstances in which a third strain, $C$, can coexist with two already coexisting strains $A$ and $B$ (Fig 2A) Note that strain $C$ has an intermediate growth rate and affinity, i.e., $g_A < g_C < g_B$ and $K_A < K_C < K_B$. Naively, one might assume that any strain whose growth parameters $g_C$ and $K_C$ lie within the shadow of the two other strains $A$ and $B$, would be able to coexist with them. However, simulations of our model with three strains revealed that this is not the case (Fig A in S1 Text); instead the region where all three strains truly coexist with each other (shown in dark grey, Fig 2B) is a smaller sub-region of the intersection of both

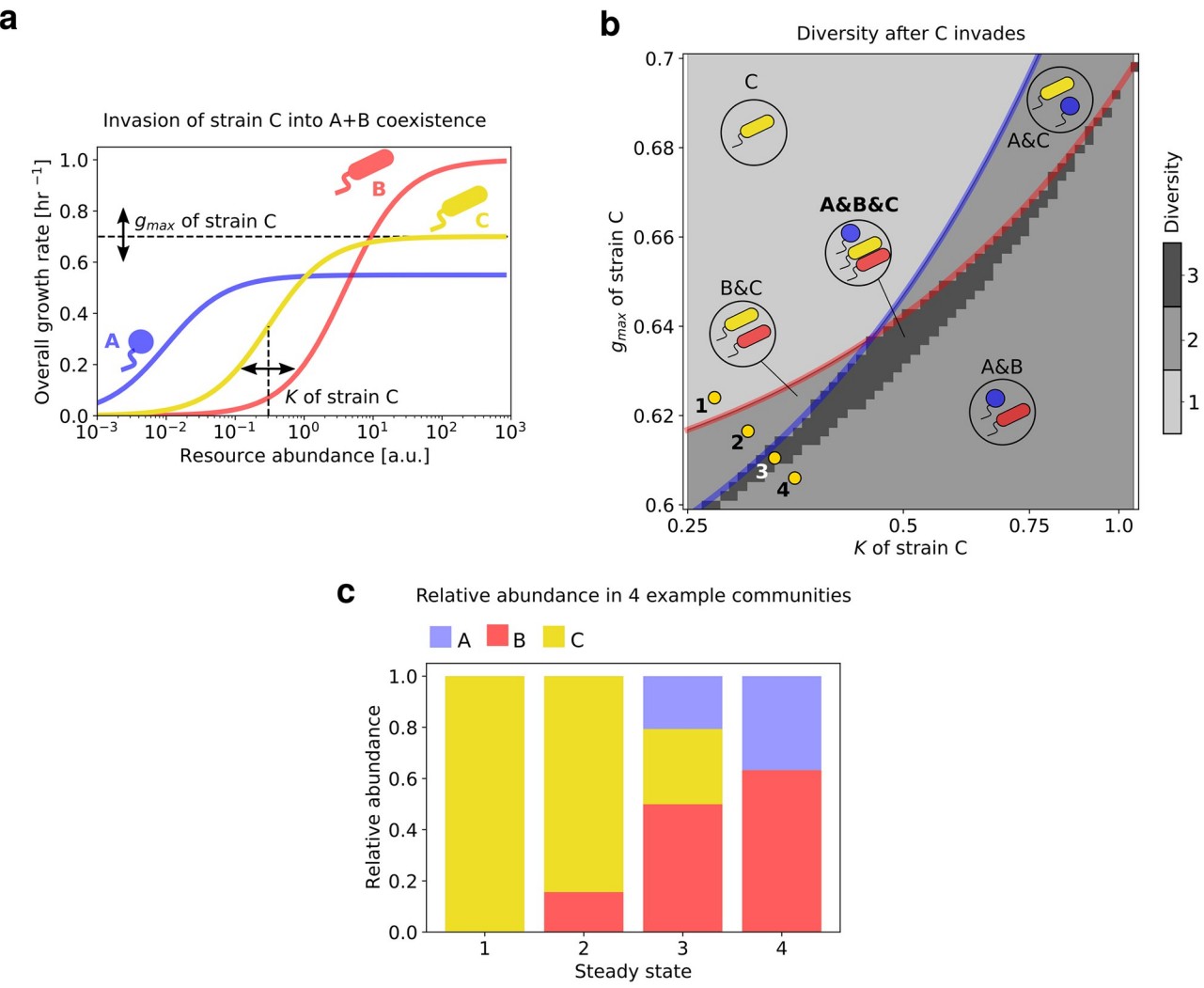

**Fig 2. Three-strain coexistence in a boom and bust environment on a single resource.** (a) Growth rate profiles of strains *A*, *B* and *C*, which all obey Monod growth. We fixed the parameters $g_{max}$ and *K* for strains *A* (blue) and *B* (red), and varied them for strain *C* in a manner that both $g_C$ and $K_C$ remained between those of strains *A* and *B*. (b) Heatmap showing the steady state community diversity (number of coexisting strains) as a function of the growth parameters $g_C$ and $K_C$. In the region above the solid blue line, *A* goes extinct, above the red line, *B* goes extinct. The yellow dots represent 4 variants of *C* corresponding to 4 qualitatively distinct steady state community compositions, shown as stacked bar plots of relative abundances in (c).

pairwise shadows (Fig A in S1 Text). In hindsight, this is not surprising. Indeed, two strains lying in the shadow of each other only ensures their pairwise coexistence. Generally, pairwise coexistence between all pairs does not guarantee that all three strains will be able to simultaneously coexist. As an illustrative example, in a chemostat model with two resources, competitive exclusion may allow strains to exist in pairs, but prohibit them from all coexisting together (since in this case, the number of strains cannot exceed the number of resources).

An important question to ask about strains occupying satellite niches is whether their abundances will be much lower than the strain with the highest maximal growth rate *g* (here, strain *A*), occupying the key resource niche. By tracking the abundance distributions of communities within and near the shadows of the three strains, we found that when multiple strains coexist, their abundances are not greatly skewed towards strain *A*, instead being comparable to each other (Fig 2C). Specifically, as we sample communities with growth parameters $g_C$ and $K_C$

across a line shown in Fig 2C, the community structure changes from consisting solely of strain C (Fig 2B, point 1), to coexistence between B and C (point 2), to all three strains coexisting (point 3), and finally to point 4, where strains A and B coexist, but C goes extinct.

To investigate how many strains can typically coexist in such environments, we simulated community dynamics of a large randomly generated pool of 100 strains (Fig 3A), whose growth parameters g and K were sampled from suitably chosen probability distributions (individual points in Fig 3C; Methods). In over 2,000 such simulations initialized with different species pools, we carried out community assembly simulations (Methods). The most common outcome in these simulations was a community with just one strain (96.6% of cases), however in 3.35% of simulations, we did observe 2 strains and in the remaining 0.05%, 3 strains. The likelihood of observing n strains decreased approximately exponentially with n. In fact, our simulations could have yielded communities with even more than 3 strains, but given the expected rarity of that occurring, it would not be a pragmatic use of computational resources. The observation that with a reasonable species pool size and number of simulations, one obtains no more than 3 strains coexisting, is in agreement with results from Stewart and Levin [22]. The surviving strains (Fig 3C, dark red points) in our three-strain communities had among the highest g and lowest K values, and appeared to lie on a Pareto frontier. However, predicting exactly which of the strains on the Pareto frontier will survive is challenging.

## Simplified model and the statistics of coexistence

In the case of strains following Monod growth, obtaining the boundaries of the region where three or more strains can coexist is mathematically challenging, because of which we resorted to numerical simulations with a large pool of strains. However, we also studied a simplified non-linear growth law where the boundaries of the coexistence region for any number of strains could be computed exactly (see S1 Text, equation S35 for 3 species and S39 for an arbitrary number of species, n). Microbial growth in this simplified version of the model approximates Monod's law by a step-like function, with the growth rate being zero below resource concentration K, and g above concentration K. Using this simplified model, we can calculate the boundaries of the coexistence region exactly, and obtain the probability of n randomly chosen strains to all coexist, where the strains have uniformly distributed $g_{max}$ and K in suitable intervals, as the following (derivation in S1 Text):

$$P_n = \frac{1}{n!}\left(\frac{D}{D-1} - \frac{1}{\log D}\right)^{n-1}, \tag{7}$$

This expression shows that the coexistence volume shrinks faster than exponentially with the number of strains that need to coexist (since $D/(D-1) - 1/\log D < 1$).

An extension of this simplified model, with multiple threshold concentrations below which the growth rate decreases, can approximate any growth law. This extension is mathematically equivalent to another model [28] where microbial strains grow in a diauxic manner, starting from their most preferred resource, switching to progressively less preferred resources, depleting one resource at a time. Unlike in diauxie, where different species can have different resource preferences, in our simplified model, the preferences of all species are identical, decreasing from the highest concentration onwards. It is due to these identical preferences that the coexistence is especially unlikely (as seen in Eq (7)) [11, 28].

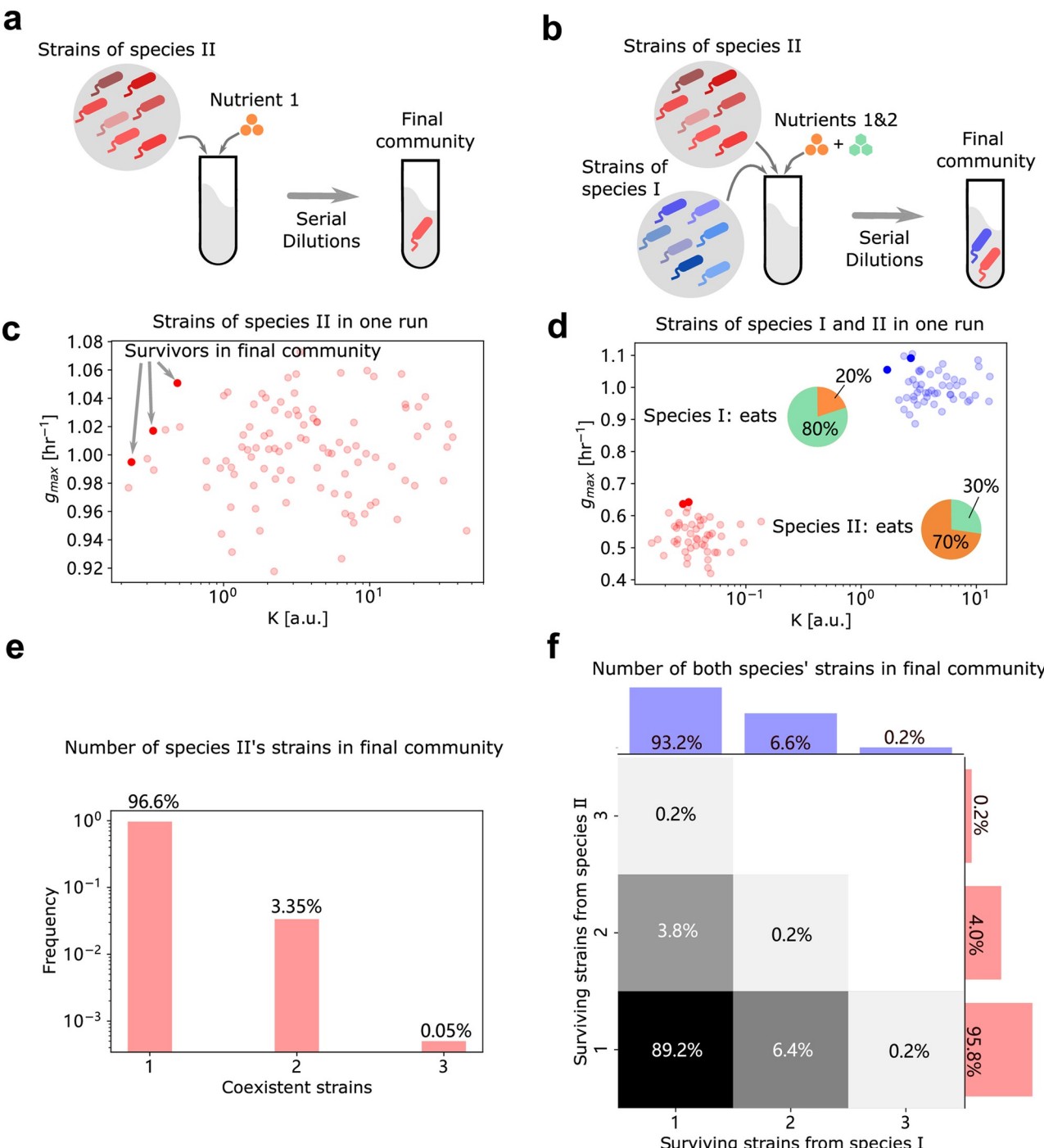

**Fig 3. Communities in multi-resource environments.** Panels (a)(c)(e) show results from single species in a single resource environment, and (b)(d)(f) depict results from 2 species in a 2-resource environment. **(a)-(b)** Schematic of strain pool simulations. All randomly generated strains from each species are presented together at the beginning of each serial dilution simulation, which will reach a final community composition at steady state. **(c)-(d)** Examples of strains in a strain pool. Dots show the $g_{max}$ and $K$'s of each strain, where the deep-colored ones represent the strains that survive in the final community. Blue dots are strains from species I and red from species II. Pie charts in (d) show the co-utilization profiles of each species. **(e)-(f)** Histograms of the number of survivors in the final community. Bar plots of blue and red show correspondingly the number of surviving strains from species I and II. Heatmap in (f) separately represents the joint distribution of the number of survivors in both species.

## Extending results to complex communities with multiple species and resources

Until now, we showed that multiple strains specializing on only one limiting resource could coexist with each other by using different $r$ or $K$ strategies. In natural communities, however, strains will not only have access to multiple resources, but also face competition from multiple other species. Indeed, different species might have evolved different resource preferences, lowering niche overlap and increasing the probability of coexistence with each other. In light of this, we next investigated whether our proposed mechanism of strain coexistence—each with different $g$ and $K$ parameters—was robust to the presence of multiple resources and species. As the simplest such extension of our previous results, we simulated community assembly of strains from two distinct species, I and II, on two different substitutable resources, $R_1$ and $R_2$. Each species co-utilized both resources, but each had a different preference towards one of the two resources: namely, species I preferred $R_1$, while species II preferred $R_2$. We also assumed that all strains of the same species had identical resource preferences, but different growth parameters, $g$ and $K$, on each of the two resources (see Methods). We then simulated community assembly via serial dilutions, each starting with a different pool of randomly generated strains belonging to two species, whose resource preferences were fixed (Fig 3C and 3d). At the end of several growth-dilution cycles, we asked how many strains of each species could coexist in the final, steady state communities.

Fig 3D shows the outcome of one such community assembly simulation, with 50 strains belonging to each species (shown as red and blue points in Fig 3D). Species I, which preferred resource $R_1$, dedicated 80% of its consumption (quantified by the growth rate coefficient $g/K$ at low resource concentration) to $R_1$ and 20% to $R_1$; species II instead dedicated 30% of its consumption to $R_1$ and 70% to $R_2$. At steady state, two strains of species I (dark blue points in Fig 3D) and two strains of species II (dark red points in Fig 3D) coexist in the community. The growth parameters $g$ and $K$ of these strains are positioned at the Pareto frontiers (upper-left corners of the $g - K$ plane) of each of their species pools.

By performing community assembly simulations over 500 independently generated species pools, we observed that in 89.2% of cases, both species could coexist, but only with one strain in each species (Fig 3F). In the remaining 10.8% of cases, at least one species was represented by >1 strain. The blue and red bars in Fig 3F show the marginal distributions of the number of strains observed in each of the two species (Fig 3F). Note that each of the marginal distributions bears striking resemblance to the histogram in Fig 3E, where we simulated only one species and one resource. Thus, our results about the coexistence of multiple strains of the same species are robust to the inclusion of more than one resource (or niche) occupied by different species.

## Methods

### Serial dilutions

In our serial dilution simulations, we initialize the population of each strain to an equal size of 1 arbitrary abundance unit. In each dilution cycle, $c_0 = 10$ units of every resource is added into the system, and strains grow according to Monod's law for $T = 24$ hr. At the end of each cycle the system is diluted by a factor of $D = 10$.

### Strain coexistence

We use two example strains (A and B) to investigate their coexistence in a single resource environment. Their maximum growth rates are arbitrarily fixed to 0.55 hr$^{-1}$ and 1.0 hr$^{-1}$

respectively, with substrate affinity being similarly fixed to 0.01 a.u. and 4.0 a.u. In Fig 2b, we introduced an invader strain C whose maximum growth rates vary between [0.6, 0.7] hr$^{-1}$ and substrate affinity vary between [0.25, 1.05] a.u.

## Strain pool simulations

For the single resource simulations, we generated 2000 strain pools of species II independently, each containing 100 mutant strains. Their maximum growth rates were sampled independently from a normal distribution of $\mathcal{N}(1.0, 0.03^2)$ in hr$^{-1}$. Their substrate affinities were sample independently from a log-normal distribution of logNormal(log 4, 0.5$^2$) in a.u.

For the 2-resource simulations, we generated 500 strain pools independently, each containing 50 mutant strains from species I and 50 from species II. Both species co-utilize the 2 resources, and as an example, the overall growth rate of strain A is

$$g_{\text{total}}^A = x^A \cdot \frac{g_{\max}^A R_1}{K_A^{(1)} + R_1} + (1 - x^A) \cdot \frac{g_{\max}^A R_2}{K_A^{(2)} + R_2}, \tag{8}$$

where $x^A$ and $1 - x^A$ are coefficients of co-utilization.

The coefficient of $x$ is fixed at 0.2 for species I and 0.7 for species II. The maximum growth rates of mutant strains of species I and II are sampled from normal distributions $\mathcal{N}(1.0, 0.03^2)$ and $\mathcal{N}(0.55, 0.03^2)$ respectively. Strains of species I have their substrate affinities sampled from log-normal distributions: $K_I^{(1)} \sim$ logNormal(log 4, 0.3$^2$), and $K_I^{(2)} \sim$ logNormal(log 5, 0.3$^2$). Strains of species II also have their substrate affinities sampled from log-normal distributions: $K_{II}^{(1)} \sim$ logNormal(log 0.05, 0.3$^2$), and $K_{II}^{(2)} \sim$ logNormal(log 0.03, 0.3$^2$).

In Fig B in S1 Text, we also investigated the case where co-utilization coefficients $x$ is variable. They are independently sampled from normal distribution, $\mathcal{N}(0.2, 0.1^2)$ for species I and $\mathcal{N}(0.7, 0.1^2)$ for species II.

## Discussion

In this paper, we showed that in time-varying environments, the competitive exclusion principle can be broken through the formation of a few satellite niches alongside the primary resource niche. These niches can be occupied by closely related strains belonging to the same species, which rapidly diversify in two key parameters, i.e., their maximum growth rate and resource affinity. Our results add to a substantial body of work investigating how nonlinearities in resource-dependent growth (i.e., Monod growth) can lead to violations of the competitive exclusion principle [21, 22, 29]. As in these studies, in our work satellite niches only appear in time-varying environments, such as the boom and bust cycles we investigated here, and completely disappear in static environments, such as chemostats. In chemostat-like static environments, resource concentrations and microbial abundances reach a fixed value at steady state, resulting in strict competitive exclusion, where only one strain can survive per resource (primary niche). The main property of time-varying environments that create satellite niches is that resource concentrations change with time, creating opportunities for different strains to specialize and be more competitive in different concentration ranges. Specifically, our work provides new mathematical expressions for the coexistence criteria applicable to two strains, connecting several relevant parameters, especially the time period between successive growth-dilution cycles, the dilution factor, as well as the growth parameters of the strains (Eqs (3)–(5)). Another new aspect of our study relative to prior work in this area, in particular Ref. [22], is the coexistence of multiple ($\geq$3) strains of each of the species in environments with more

than one supplied resource (primary niches). It is reasonable to expect that strains could modify the ratio in which they consume different resources, i.e., change how they allocate their enzyme budgets. Following in the footsteps of Good et al. [30], we adapted our model and extended it to include the possibility of small variations in resource budget allocation by closely related strains. We found that variation in budget allocation alone rarely (1% of simulations) leads to coexistence of multiple strains, compared with variation in growth parameters alone (11.8% of simulations, Fig B in S1 Text). Thus, satellite niches created by boom and bust cycles chiefly select for variation in such growth parameters.

The existence of organisms which specialize either on rapid growth (*r*-strategists) or more complete depletion of the available resources (*K*-strategists) is well-established in natural ecosystems [24, 25, 31]. Examples include microbes residing in Earth's upper ocean, where *r*-strategists dominate in strongly seasonal temperate oceans, while *K*-strategists dominate in stable, low-seasonality, equatorial marine environments [32, 33]. However, whether such *r*- and *K*-strategists can coexist in the same environment has not been fully explored. Our results show that *r*- and *K*-strategists may indeed coexist as closely related strains of the same species, both in simple and complex multi-species, multi-resource environments. These predictions also match spatial distributions of strain diversity of microbial populations in Earth's upper oceans predicted in ref. [34]. While the strain diversity in highly seasonal environments (up to 8) was somewhat higher than predicted by our model (up to 3), the authors readily admit that their predictions might be somewhat inflated by ocean dynamics, transiently mixing organisms from different habitats. Furthermore, rapid and continuous evolution might also increase the apparent diversity of a community, due to transient strains generated by mutations and ultimately lost due to competitive exclusion. Our model included a fixed pool of strains and provided sufficient time to achieve a steady state as a result of competitive exclusion, and thus ignored the transient diversity due to rapid evolution.

Our work intriguingly suggests that a trade-off between Monod growth parameters ($g_{max}$ and $1/K$) promotes coexistence, and thus might be observed among coexisting microbial strains in natural communities. Past work [35, 36] has indicated similar trade-offs between maximal growth rate and efficiency, or yield (the fraction of environmental carbon converted to biomass), some even suggesting that these trade-offs become more pronounced at lower taxonomic levels [35]. While the yield or efficiency parameter in these studies does not influence two-strain coexistence in our model (S1 Text), a trade-off between maximal growth rate ($g_{max}$) and affinity ($1/K$) in our model has been discussed and reviewed in Ref. [37], suggesting that they might promote coexistence in natural bacterial and phytoplankton communities. Future work examining such a trade-off and its causal implication for coexistence would be fruitful.

Another natural environment in which limited strain diversity may have been detected is the human gut microbiome [7], with a previous study reporting the coexistence of a few strains (between 1 and 3, termed "oligo-colonization"). In particular, the authors mention that "it is not clear what mechanisms would allow for a second or third strain to reach intermediate frequency, while preventing a large number of other lineages from entering and growing to detectable levels at the same time". Our work, in quantitative agreement with these observations, suggests that a possible mechanism explaining them might be that closely related strains differ in their growth parameters ($g_{max}$ and $K$) by which they consume resources.

"Oligo-colonization" is not limited to natural environments such as ocean and gut microbiomes, but also likely manifests in microbial communities domesticated in the lab [16]. In these domesticated communities, between 1 and 2 closely related strains of the same species were found to coexist over multiple ($\sim 70$) boom and bust (serial dilution) cycles, consistent with the predictions of our model. Thus, taken together, our model explains a possible

mechanism by which such "oligo-colonization" of a small number of closely related strains might be widespread in both natural and laboratory-domesticated microbial communities.

## Supporting information

**S1 Text. Supplementary Text, including derivations, figures containing additional results, and detailed calculations for the simplified model. Fig A. A detailed view of multi-strain coexistence, similar to** Fig 2B. Heatmaps showing the final community diversity (number of strains) as a function of the growth parameters $g_C$ and $K_C$. Given a strain C with variable $g_{max}$ and $K$ along with fixed strains A and B, the heatmap shows the number of coexisting species in the steady state of serial dilutions. Here, the region above the blue line represents where C drives A out of the steady state, and the region above the red line represents where B is driven out by C. The green regions represent areas where more than one strain can coexist. (b) shows a zoomed-in view of the boxed region in (a), highlighting the region where two (light green) and three (dark green) strains can coexist. **Fig B. Histograms showing statistics of the number of coexisting strains**: (a) when we allow small variations in how a species partitions two resources, with no variation in Monod growth parameters, and (b) when we allow changes in both the Monod parameters as well as resource partitioning. Bar plots on the margins in blue and red show the number of surviving strains from species I and II respectively. In (a), the variation in resource partitioning was as follows: each species' resource partitioning was independently sampled from a normal distribution, $\mathcal{N}(0.2, 0.1^2)$ for species I and $\mathcal{N}(0.7, 0.1^2)$ for species II. **Fig C. Growth curve of the simplified model**. The 3 solid lines show how growth rate depends on the resource concentration in our simplified model. Here $g_A$, $g_B$ and $g_C$ are approximations of the maximum growth rate from Monod's law of growth, and $K_A$, $K_B$ and $K_C$ are approximations of the substrate affinity. **Fig D. Monod growth curve approximated by the simplified model**. The grey line shows the growth curve of a strain that obeys Monod's law of growth on a single resource, with maximum growth rate $g_A^{Monod}$ and affinity $K_A^{Monod}$. The blue line is the growth growth curve of a strain that grows on a set of unique resources and follows our simplified growth model with growth rates $g_A^{(1)}, g_A^{(2)}$... etc. and affinity $K_A^{(1)}, K_A^{(2)}$... etc. **Fig E. 2-dimensional vector space representing steady state of a serial dilution process**. Horizontal axis serves to measure the species abundances at the point $c = K_1$ while vertical axis measures species abundance at the point $c = K_2$ scaled to some appropriate unit of abundance. Vectors $\vec{B}_1$ and $\vec{B}_2$ represent species $B_1$ and $B_2$ respectively, their coordinates proportional to the abundances of these species at the points $c = K_1$ (first coordinate) and $c = K_2$ (second coordinate) respectively. The blue-shaded area consists of points representing positive linear combinations of vectors $\vec{B}_1, \vec{B}_2$. A red point within the blue-shaded area, with the parameters $K_1$ and $K_2$ corresponding to it (shown in red color) provides the values $K_1$, $K_2$ favoring coexistence of two species with given growth rates $g_1, g_2$ in the given environment (as any other point in the blue-shaded area does). Coordinates of vectors $\vec{B}_1, \vec{B}_2$ given in dimensionless numbers are to be interpreted as number×unit of abundance.
(PDF)

## Acknowledgments

We thank Maria Verbitsky for writing the Python code to find the environmental conditions in which any two strains may coexist. We thank the Kavli Institute for Theoretical Physics, UCSB, where this project took root, and Daniel Fisher for discussions which helped lead us to the analyses in this paper.

## Author Contributions

**Conceptualization:** Sergei Maslov, Akshit Goyal.

**Formal analysis:** Yulia Fridman, Zihan Wang, Akshit Goyal.

**Funding acquisition:** Sergei Maslov.

**Investigation:** Yulia Fridman, Sergei Maslov, Akshit Goyal.

**Methodology:** Yulia Fridman, Zihan Wang, Sergei Maslov.

**Project administration:** Sergei Maslov.

**Software:** Zihan Wang.

**Supervision:** Sergei Maslov, Akshit Goyal.

**Visualization:** Yulia Fridman, Zihan Wang, Sergei Maslov, Akshit Goyal.

**Writing – original draft:** Yulia Fridman, Sergei Maslov, Akshit Goyal.

**Writing – review & editing:** Yulia Fridman, Sergei Maslov, Akshit Goyal.

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
