## [Decision Letter · Decision Letter 0]

15 Aug 2022

Dear Goyal,

Thank you very much for submitting your manuscript "Fine-scale diversity of microbial communities due to satellite niches in boom-and-bust environments" for consideration at PLOS Computational Biology.

As with all papers reviewed by the journal, your manuscript was reviewed by members of the editorial board and by several independent reviewers. In light of the reviews (below this email), we would like to invite the resubmission of a significantly-revised version that takes into account the reviewers' comments.

We cannot make any decision about publication until we have seen the revised manuscript and your response to the reviewers' comments. Your revised manuscript is also likely to be sent to reviewers for further evaluation.

Sincerely,

Jacopo Grilli

Associate Editor

PLOS Computational Biology

Ville Mustonen

Deputy Editor

PLOS Computational Biology

Reviewer's Responses to Questions

**Comments to the Authors:**

Reviewer #1: The review is uploaded as an attachment.

Reviewer #2: See attachment

**Have the authors made all data and (if applicable) computational code underlying the findings in their manuscript fully available?**

Reviewer #1: Yes

Reviewer #2: Yes

PLOS authors have the option to publish the peer review history of their article (what does this mean?). If published, this will include your full peer review and any attached files.

Reviewer #1: No

Reviewer #2: No
---

## [Decision Letter · Decision Letter 1]

5 Dec 2022

Dear Goyal,

We are pleased to inform you that your manuscript 'Fine-scale diversity of microbial communities due to satellite niches in boom and bust environments' has been provisionally accepted for publication in PLOS Computational Biology.

Best regards,

Jacopo Grilli

Academic Editor

PLOS Computational Biology

Ville Mustonen

Section Editor

PLOS Computational Biology

Reviewer's Responses to Questions

**Comments to the Authors: **

Reviewer #1: The authors did an excellent job both refining and expanding their manuscript. The new analyses on the role of yield, effectively finite dilution, and phase volume were particularly informative. I have no additional comments and believe that the manuscript should be accepted.

Reviewer #2: I feel that the authors have addressed all the comments and concerns in my last review. I recommend publication.

**Have the authors made all data and (if applicable) computational code underlying the findings in their manuscript fully available?**

Reviewer #1: Yes

Reviewer #2: Yes

PLOS authors have the option to publish the peer review history of their article (what does this mean?). If published, this will include your full peer review and any attached files.

Reviewer #1: **Yes: **William R. Shoemaker

Reviewer #2: No

---

## [Editor Report · Acceptance letter]

20 Dec 2022

PCOMPBIOL-D-22-00805R1 

Fine-scale diversity of microbial communities due to satellite niches in boom and bust environments

Dear Dr Goyal,

I am pleased to inform you that your manuscript has been formally accepted for publication in PLOS Computational Biology. Your manuscript is now with our production department and you will be notified of the publication date in due course.

With kind regards,

Zsofi Zombor
